# RALR: Random Amplify Learning Rates for Training Neural Networks

Jiali Deng [1,2], Haigang Gong [1,2], Minghui Liu [1,2], Tianshu Xie [1], Xuan Cheng [1,2], Xiaomin Wang [1,2] and Ming Liu [2,*]

1   School of Computer Science and Engineering, University of Electronic Science and Technology of China, Chengdu 611731, China; dengjiali@std.uestc.edu.cn (J.D.); hggong@uestc.edu.cn (H.G.); minghuiliuuestc@163.com (M.L.); tianshuxie@std.uestc.edu.cn (T.X.); cs_xuancheng@std.uestc.edu.cn (X.C.); xmwang@uestc.edu.cn (X.W.)
2   Yangtze Delta Region Institution (Quzhou), University of Electronic Science and Technology of China, Quzhou 324000, China
*   Correspondence: csmliu@uestc.edu.cn

**Abstract:** It has been shown that the learning rate is one of the most critical hyper-parameters for the overall performance of deep neural networks. In this paper, we propose a new method for setting the global learning rate, named random amplify learning rates (RALR), to improve the performance of any optimizer in training deep neural networks. Instead of monotonically decreasing the learning rate, we expect to escape saddle points or local minima by amplifying the learning rate between reasonable boundary values based on a given probability. Training with RALR rather than conventionally decreasing the learning rate achieves further improvement on networks' performance without extra consumption. Remarkably, the RALR is complementary with state-of-the-art data augmentation and regularization methods. Besides, we empirically study its performance on image classification tasks, fine-grained classification tasks, object detection tasks, and machine translation tasks. Experiments demonstrate that RALR can bring a notable improvement while preventing overfitting when training deep neural networks. For example, the classification accuracy of ResNet-110 trained on the CIFAR-100 dataset using RALR achieves a 1.34% gain compared with ResNet-110 trained traditionally.

**Keywords:** RALR; learning rate; saddle points; deep neural networks

## 1. Introduction

After decades of lower activity, studies in artificial neural networks were revitalized after a 2006 breakthrough [1] in the research on deep learning. Deep neural networks have been a tremendous practical success in many fields, such as image recognition [2], object detection [3], face recognition [4], object tracking [5], and speech recognition [6]. However, a deep neural network is a complex mathematical function, consisting of millions of parameters that need to be computed to solve a problem. In addition, the parameters of deep neural networks include weight parameters and hyper-parameters. Each hyper-parameter has its impact on training deep neural networks.

Deep neural networks are usually learned to perform a specific task, and the goal during training is to minimize the objective function $f = \mathbb{R}^n \to \mathbb{R}$, typically updated by stochastic gradient descent (SGD). The basic procedure to optimize $f$ is:

$$\theta_{t+1} = \theta_t - \eta_t \nabla f_t(\theta_t) \tag{1}$$

Here, $\theta_t \in \mathbb{R}^n$ is the weight parameter vector at the $t$-th iteration, $\nabla f_t(\theta_t)$ is the loss gradient information obtained on a relatively small batch of training datasets at the $t$-th iteration, and $\eta_t$ is the learning rate at the $t$-th iteration.

According to Equation (1), we perceive that the learning rate is a crucial hyperparameter that controls how much we are adjusting the weight parameter vector of deep neural networks concerning the loss gradient information obtained by training datasets. It is well known that a small learning rate will make the training procedure converge slowly and more prone to get stuck in saddle points or local minima, and a large learning rate will cause the training procedure to diverge.

Traditionally, the learning rate should be a single value that monotonically decreases during training at specific epochs. This paper describes the remarkable phenomenon that is randomly amplifying the learning rate during training is beneficial overall. Hence, we propose randomly magnifying the global learning rate between reasonable boundary values based on the traditional monotonically decaying way, named random amplify learning rates (RALR). The RALR effectively makes the networks' performance near the optimal classification accuracy compared to the conventional monotonically decaying way. Furthermore, unlike the adaptive learning rates methods, the RALR procedure requires virtually no additional computation.

To better understand the potential profits of RALR, we visually plot the training and validation loss, validation accuracy, and the global learning rate changing of both our RALR method and traditional method while training the CIFAR-100 dataset using network ResNet-110 in Figure 1. The blue curve in Figure 1b shows the baseline of ResNet-110p; the baseline of ResNet-110 reaches the best classification accuracy of 73.94% after 300 epochs. In contrast, it is possible to achieve 2.11% gain in the same epochs using the RALR technique, and the orange curve in Figure 1 shows the validation accuracy while training using RALR, the network ResNet-110 reaches the best classification accuracy of 76.05% at the 291st epoch. In Figure 1a, we found that training with traditional strictly decreasing methods (blue line) is more prone to overfitting compared with training using RALR (orange line). As shown in Figure 1, enlarging the global learning rate may temporarily reduce the classification accuracy and increase the training and verification loss. Still, the classification accuracy perhaps reaches a higher level after the global learning rate is restored.

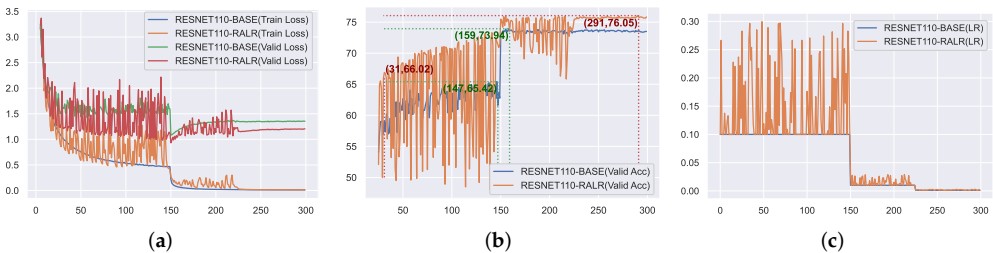

**Figure 1.** Training and validation loss, validation accuracy, and changing of learning rate while training the CIFAR-100 dataset using network ResNet-110. (**b**) The orange curve shows the validation accuracy of the new learning rate policy. (**a**) Loss. (**b**) Validation accuracy. (**c**) Learning rate.

The contributions of this paper are:

1. A methodology for setting the global learning rates for training deep neural networks makes the performance near-optimal classification accuracy compared with traditional monotonically decaying methods, with essentially no additional computation. Numerical and graphical results show that our method outperforms the traditional learning rate decreasing way.
2. RALR is applicable to any optimizers, such as SGD with Nesterov momentum [7], adaptive gradient (Adagrad [8]), and adaptive moment estimation (Adam [9]).
3. RALR is complementary to existing data augmentation and regularization techniques, such as Mixup [10], Cutout [11], CutMix [12], and Label Smoothing [13].
4. RALR is demonstrated with various architectures on image classification tasks, fine-grained classification tasks, object detection tasks, and machine translation tasks.

## 2. Motivation

Training deep neural networks involves using an optimization algorithm to find the weight parameter vector to best map inputs and outputs. Many researchers discover how to accelerate the learning speed, reduce overfitting, and obtain optimal networks' performance with deep learning models. The training procedure is challenging, not least because the optimization function is non-convex and can be falling into the local minima or saddle points problems, as shown in Figure 1b. Training with conventionally decaying way cannot receive better performance after 159 epochs (blue line). Yet, training with RALR is less prone to overfitting, which improves the networks' performance continuously almost in the whole training process (orange line), and the network gains the best classification accuracy at the 291st epoch. The training procedure of RALR is more effective and can speed up convergence ability and mitigate against overfitting (see Figure 1b). The validation accuracy of the baseline reaches the highest 65.25% at the 139th epoch in the first stage. In comparison, the validation accuracy of RALR achieves 66.02% at the 31st epoch and reaches the highest 73.30% at the 142nd epoch in the first stage.

The global learning rate is a hyper-parameter that controls how much to change the model. Traditionally, the global learning rate decreases at specific epochs, i.e., the learning rate may become smaller and smaller, and the learning rate might be too small after some epochs, resulting in the optimizer sticking in saddle points or local minima that may slow down the training speed and restrict the network performance in a long time [14]. The x-axis on each subfigure of Figure 2 indicates the value of the activation units, and the y-axis indicates the number of activation units under this specific value. As shown in Figure 2a, the activation values of test images or the network parameters are substantially unchanged for the last 100 epochs, which indicates, the network cannot update effectively in the remaining training time. Increasing the learning rate between reasonable boundary values at the specific probability can potentially help the training escape saddle points and local minima and find flatter minima with good generalization performance by the leap forward of the network's weights updates. Figure 2b shows that the network's activation values and weight parameters are updated effectively with the RALR for the last 100 epochs. That means the network may restructure the parameter weight vector to gain near-optimal performance. Each column of Figure 2 indicates the activation values of different convolutional layers for one given image. The first, second, and last column indicate the activation values of the layer1, layer2, and layer3 of ResNet-110.

From the above discussion, if the training procedure does not improve the network performance anymore, we can amplify the learning rate between reasonable boundary values to accelerate the network's training and achieve better performance. Increasing the learning rate allows the optimizer of deep neural networks to escape saddle points and local minima effectively. In this way, it is possible to further explore the optimization algorithm and leads the network to better generalization, as illustrated in Figure 2.

We propose the RALR method to alter the learning rate around the global one. Amplifying the learning rate at some epochs can help the optimization algorithm escape local minima or saddle points and quickly find the correct descent direction. Particularly, the larger learning rate will make the optimizer explore the larger hidden space, which indicates the network could be updated more effectively. As a result, we speed the training up and train the network more effectively in comparison with the previous one.

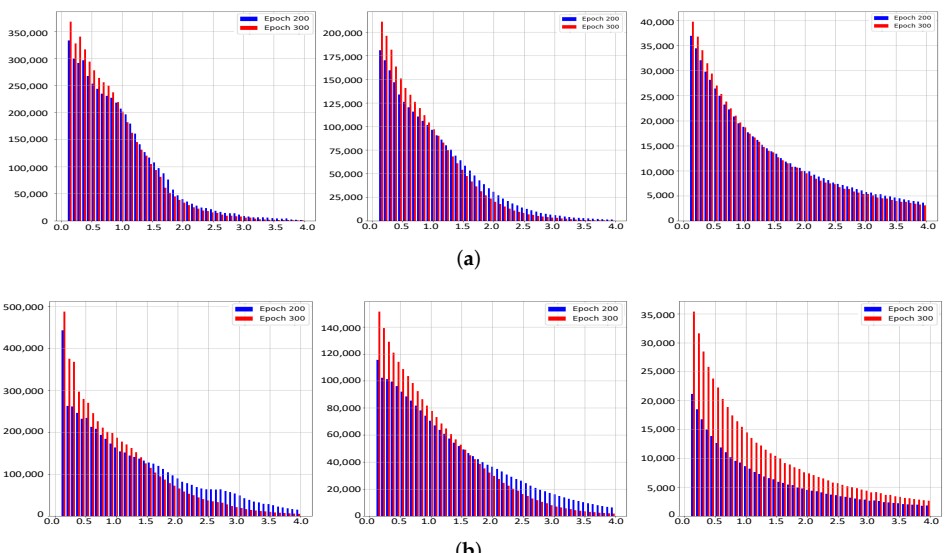

**Figure 2.** The activation values changing of a network ResNet-110 trained for 300 epochs. The blue column indicates activation values of the network ResNet-110 obtained at the 200th epoch, and the red column indicates activation values of the network ResNet-110 obtained at the 300th epoch. (**a**) Baseline. (**b**) RALR.

## 3. Related Works

Training deep neural networks is a challenging optimization task. Many researchers try to achieve better network performance and accelerate the training speed. Some works are described briefly in the section below.

Adaptive Learning Rates: In order to improve the convergence speed of gradient descent learning algorithms, momentum [7] is proposed to accelerate SGD in the relevant direction. Instead of utilizing the gradient of the current step individually to guide the direction, momentum also accumulates the gradient of the previous step. As one of the early adaptive methods, AdaGrad [8] is a dynamic learning rate decreasing algorithm. AdaGrad adapts different learning rates associated with different features, i.e., frequently occurring features updated using small learning rates and infrequently occurring features updated using large learning rates. RMSprop is introduced by Geoffrey Hinton in Lecture 6e of his Coursera Class (http://www.cs.toronto.edu/~tijmen/csc321/slides/lecture_slides_lec6.pdf, accessed on 25 December 2021). RMSprop divides the learning rate by an exponentially decaying average of squared gradients. AMSGrad [15] pinpoints the exponential moving average of past squared gradients as a reason for the poor generalization behavior of earlier adaptive learning rate methods and uses the maximum of past squared gradients rather than the exponential average to update the parameters.

In recent years, several optimizers have been proposed. These include AdamW [16], which fixes weight decay in Adam; QHAdam [17], which averages a standard SGD step with a momentum SGD step; and AggMo [18], which combines multiple momentum terms; and others.

Cycling learning rates: CLR and SGDR: The idea of cycling increasing the global learning rate has been explored by algorithms such as SGDR [19] and CLR [20]. SGDR reset the global learning rate to a large value in a periodic manner. After this, the global learning rate declines to a small value following a cosine annealing schedule. In addition, CLR lets the global learning rate cyclically vary between reasonable boundary values. Leslie N. Smith [20] argued that we could estimate a reasonable learning rate by initially training the model with a very small learning rate and increasing the global learning rate linearly or exponentially at each iteration in CLR.

Unlike the above techniques, RALR amplifies the global learning rate based on the given probability. It prevents the model from overfitting by escaping saddle points or

local minima. Further experiments demonstrate the effectiveness of the RALR in achieving better performance.

Others: Biler et al. [21] stressed that the learning rate of the gradient descent strongly affects performance in neural networks and propose the Alrao algorithm. Blier et al. believed that each unit or feature in the network gets its learning rate sampled from a random distribution spanning several orders of magnitude, hoping that enough units will get a close-to-optimal learning rate.

Unlike Alrao (the experimental results of Alrao show that Alrao cannot effectively improve the network performance on CIFAR and ImageNet datasets. Besides, the code of Alrao is not publicly available. Therefore, we are not able to provide the comparison results with ALRAO here), RALR is easy to implement and takes the same learning rate for all units or features of the network.

## 4. Method

### 4.1. Observation

From Equation (1), we have known that the learning rate is a crucial hyper-parameter for SGD. To characterize the surprising phenomenon of RALR, we use the class activation mapping (CAM) [22] to visualize the activation units of network ResNet-110 trained on the Tiny-ImageNet dataset (http://tiny-imagenet.herokuapp.com/, accessed on 25 December 2021), and a class activation map for a particular category indicates the discriminative image regions used by the convolutional neural network (CNN) to identify that category. Figure 3 shows the comparison of class activation mapping, where objects of RALR are more precise and more accurate than the original ones.

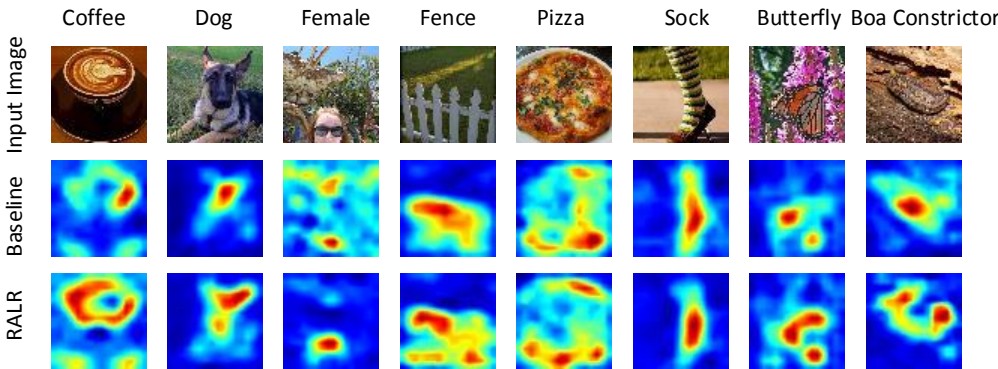

**Figure 3.** The first line shows the part of the original images of the Tiny-ImageNet test dataset. The second line shows the localized class-discriminative regions with ResNet-110 training with the traditional way. The third line shows the localized class-discriminative regions with ResNet-110 training with RALR.

### 4.2. Schedule of RALR

RALR is conducted with a certain probability of $p_r$ in the training procedure, and we name this probability factor $p_r$ as the amplification probability factor. For a training epoch $e$, the probability of it undergoing learning rate amplification is $p_r$, and the probability of it being kept unchanged is $1 - p_r$. In this process, training epochs with RALR are generated.

The procedure of RALR for training deep neural networks is shown in Algorithm 1. RALR usually sets a random boundary interval $[low, high]$, the amplification factor is set to $p_r$. Assume our training procedure takes a new training epoch $e$, $e$ is a natural number, we randomly initialize the probability $p_e$ for this epoch, if $p_e < p_r$, we amplify the learning rate with a random numerical value $random\_rate \in [low, high]$, i.e., the learning rate $\eta_e$ of this epoch $e$ is $\eta_e = \eta_o \times random\_rate$, $\eta_o$ is the global learning rate according to conventionally decaying way. Otherwise, we use the original global learning rate $\eta_o$ in this epoch for training, i.e., $\eta_e = \eta_o$.

---

**Algorithm 1** Schedule of RALR based on epoch for training deep neural networks.

---

**Require:** Model $\theta$, $N$ Training Samples $(x_i, y_i)_{i=1}^N$, Optimizer SGD, Momentum, Weight decay, Batch Size, Number of epochs $e$, Decrease learning rate epochs $e_d$, Loss Function $f(\theta)$. Initial learning rate $\eta_0$, Gamma $\gamma$.
**Require:** Random interval factor $[low, high]$, Amplification probability $p_r$
**Ensure:** Initialize $\theta, \eta_0$, best Loss $L^* \longleftarrow f(\theta)$

  1: **for** $i \in [1, e]$ **do**
  2:      **if** $i \in e_d$ **then**
  3:         $\eta_0 * = \gamma$
  4:      **end if**
  5:      Initialize random probability for each epoch $p_i$
  6:      **if** $p_i \leq p_r$ **then**
  7:         Amplification rate $random\_rate \in [low, high]$
  8:         Amplify learning rate $\eta = \eta_0 \times random\_rate$
  9:         Train for 1 epoch.
10:         Evaluate new loss $L \longleftarrow f(\theta)$
11:      **else**
12:         learning rate $\eta = \eta_0$
13:         Train for 1 epoch.
14:         Evaluate new loss $L \longleftarrow f(\theta)$
15:      **end if**
16:      **if** $L \leq L^*$ **then**
17:         Update $L^* = L$. Save checkpoint $\theta$ and $L^*$
18:      **end if**
19: **end for**

---

## 5. Experiment

This section will demonstrate the RALR schedule's effectiveness on various convolutional networks for some computer vision tasks. If not specified, the learning rate is randomly amplified at the probability of $p_r = 0.5$, the random interval factor is set to $low = 1$, $high = 3$ All experimental results are averaged over four runs and we report the standard deviation. Moreover, this section analyzes the amplification probability $p_r$ and random interval boundary values $low$ and $high$. All of the experiments are performed with PyTorch [23] on Tesla M40 or GeForce GTX 1080 GPUs.

### 5.1. RALR on Different Tasks

CIFAR-10 and CIFAR-100: Both of the CIFAR datasets [24] consist of 60,000, $32 \times 32$ color images. CIFAR-10 has ten classes, such as dog, cat, car, and boat, with 6000 images per class, including 5000 training images per class and 1000 validation images per class. There are 50,000 training images and 10,000 validation images in total. The CIFAR-100 dataset is similar the CIFAR-10, except it has 100 classes containing 600 images each. There are 500 training images and 100 validation images per class. CIFAR-100 requires much more fine-grained recognition compared to CIFAR-10, as some classes are very visually similar. For example, it contains five different species of flowers: orchids, poppies, roses, sunflowers, and tulips.

In the experiment on the CIFAR dataset, the images are first zero-padded with 4 pixels on each side to get the $40 \times 40$ pixel images. Then, $32 \times 32$ pixel images are cropped randomly. Besides, images are randomly mirrored horizontally at a probability of $p = 0.5$.

We compare the classification accuracy of some different architectures on the CIFAR-10 dataset, including deep residual networks (ResNet) [25], dense network(DenseNet) [26], and wide residual network (WRN) [27]. The networks includs (1) ResNet-56 and ResNet-110: a ResNet with a depth of 56 and 110; (2) DenseNet-BC-100-12: a DenseNet with a depth of 100 and a growth rate of 12; and (3) WRN-28-10, a WRN with a depth of 28 and a widening factor of 10.

We first show the feature embedding in Figure 4, which is the results of network ResNet-110 updated by SGD and trained on the CIFAR-10 dataset with the cross-entropy loss. Compare with the baseline one, the model trained using the RALR technique brings stronger intra-class compactness and larger inter-class separability. This means our RALR strategy can efficiently improve the network performance.

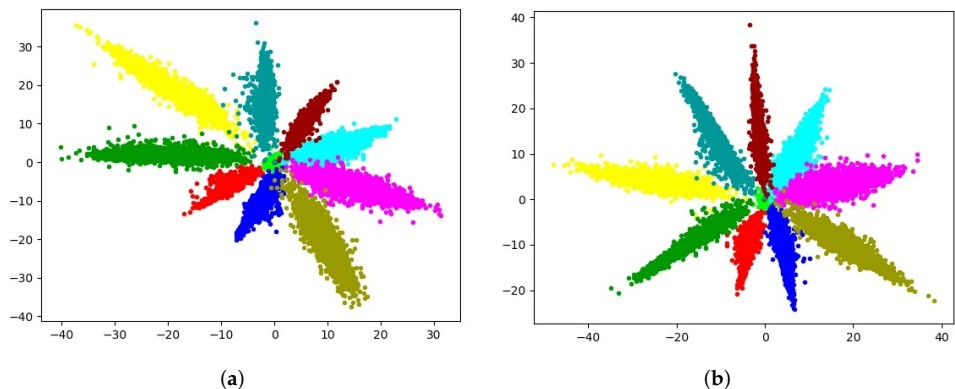

(**a**)  (**b**)

**Figure 4.** The feature embedding of ResNet-110 on the CIFAR-10 dataset. The network is trained with the cross-entropy loss and updated by SGD. Each color on the picture represents a category of the CIFAR-10 dataset. (**a**) Baseline. (**b**) Random learning rates.

We trained for 300 epochs on the CIFAR-10 dataset for the architectures listed in Table 1. For all architectures, we trained with batches of 128 images using SGD. The hyperparameter of Nesterov momentum is set to 0.9, weight decay is $1 \times 10^{-4}$, and the global learning rate is initially set to 0.1 and decreased by $10\times$ at the 150th and 225th epoch. In Table 1, we observe that the RALR technique consistently outperforms the baseline scheme.

**Table 1.** Validation accuracy (%) over different architectures on CIFAR-10.

| Model | Baseline | RALR |
|---|---|---|
| ResNet-20 | $91.41 \pm 0.08$ | $\mathbf{92.28 \pm 0.05}$ |
| ResNet-56 | $93.77 \pm 0.06$ | $\mathbf{94.35 \pm 0.05}$ |
| ResNet-110 | $94.42 \pm 0.06$ | $\mathbf{95.05 \pm 0.04}$ |
| WRN-28-10 | $95.82 \pm 0.03$ | $\mathbf{96.36 \pm 0.03}$ |

We trained for 300 epochs on the CIFAR-100 dataset for architectures listed in Table 2. For ResNets and WRNs, we trained with batches of 128 images using SGD, and we trained with batches of 64 images using SGD for DenseNet. The hyper-parameter of Nesterov momentum is set to 0.9, weight decay is $1 \times 10^{-4}$, and the global learning rate is initially set to 0.1 and decreased by $10\times$ at the 150th and 225th epoch for all architectures. In Table 2, we observe that the RALR technique consistently outperforms the baseline scheme. For example, our approach achieves 75.87% accuracy when applying network ResNet-110, which is a 1.34% improvement over baseline.

**Table 2.** Validation accuracy (%) over different architectures on CIFAR-100.

| Model | Baseline | RALR |
|---|---|---|
| ResNet-20 | $69.88 \pm 0.06$ | $\mathbf{70.48 \pm 0.03}$ |
| ResNet-56 | $73.69 \pm 0.07$ | $\mathbf{74.54 \pm 0.04}$ |
| ResNet-110 | $74.53 \pm 0.41$ | $\mathbf{75.87 \pm 0.07}$ |
| WRN-28-10 | $78.81 \pm 0.24$ | $\mathbf{79.54 \pm 0.11}$ |
| DesNet-BC-100-12 | $77.22 \pm 0.15$ | $\mathbf{77.95 \pm 0.04}$ |

Fine-grained classification: The fine-grained image classification task focuses on differentiating between hard to distinguish object classes, such as species of birds, flowers, or animals, and identifying the makes or models of vehicles. Since collecting fine-grained samples often requires expert-level domain knowledge, most fine-grained datasets are hard to extend to a large scale, i.e., the fine-grained datasets are usually insufficient to gain reliable classification accuracy of deep neural networks when adopting the random initialization method. In the fine-grained classification tasks, the deep learning models are always fine-tuned on the parameters trained on ImageNet.

We test the effectiveness of RALR on several fine-grained benchmark datasets: CUB-200-2011 [28], StandfordDogs [29], and StandfordCars [30]. In the experiments, we evaluate the performance of the fine-grained classification tasks above using ResNet-50 pre-trained on ImageNet, and the model is fine-tuned by the optimizer SGD. 'The model is trained for 95 epochs, and the learning rate is initially set to 0.001 and decayed at 30th, 60th, and 90th epoch. The comparison result is shown in Table 3. For all the cases in Table 3, models trained using RALR achieve significant boosts over baseline.

**Table 3.** Validation accuracy (%) of ResNet-50 on different fine-grained tasks.

| | CUB | | Stanford Cars | | Stanford Dogs | |
|---|---|---|---|---|---|---|
| | **+RALR** | **Acc.** | **+RALR** | **Acc.** | **+RALR** | **Acc.** |
| RseNet-50 | $\times$ | $85.32 \pm 0.23$ | $\times$ | $90.18 \pm 0.18$ | $\times$ | $84.15 \pm 0.25$ |
| | $\checkmark$ | $\mathbf{86.36 \pm 0.08}$ | $\checkmark$ | $\mathbf{90.83 \pm 0.05}$ | $\checkmark$ | $\mathbf{84.72 \pm 0.13}$ |

Object Detection: Object detection is a task to predict the bounding boxes of each object of an image, with associated real-value confidence. SSD [31] is one object detection architecture. In this section, we use SSD (https://github.com/sgrvinod/a-PyTorch-Tutorial-to-Object-Detection, accessed on 25 December 2021) to evaluate the effectiveness of RALR on object detection. For comparison, the backbone of our experiments is VGG-16 and using VOC 2007 and VOC 2012 as the training dataset simultaneously. The results are shown in Table 4. RALR can achieve further detection performance compared with the baseline one even trained with the network VGG-16 pre-trained on ImageNet-2012.

**Table 4.** Comparison results of SSD on PASCAL VOC 2012.

| **Backbone** | **+RALR** | **Acc.** |
|---|---|---|
| SSD-VGG16 | $\times$ | 77.0 |
| | $\checkmark$ | **77.7** |

Machine Translation - WMT 2014: We evaluate the capability of RALR technique on the standard WMT 2014 English–German dataset consisting of about 4.5 million sentence pairs. Each training batch contained a set of sentence pairs containing approximately 25,000 source tokens and 25,000 target tokens. The backbone is from the paper [32]. Table 5 shows the comparison results (https://github.com/sgrvinod/a-PyTorch-Tutorial-to-Machine-Translation/, accessed on 25 December 2021). The results in Table 5 show that the RALR technique outperforms the baseline and even uses the factor $low = 1$, $high = 3$, and $p_r = 0.5$ which gained in computer vision tasks.

**Table 5.** Comparison results of WMT 2014.

| Tokenization | Cased | BLEU | |
| | | Baseline | +RALR |
|---|---|---|---|
| 13a | Yes | 25.1 | **25.5** |
| | No | 25.6 | **25.9** |
| International | Yes | 25.9 | **26.1** |
| | No | 26.3 | **26.7** |

*5.2. RALR on Different Optimizers*

To verify the generalization of our proposed method, we do some experiments with ResNet-110 on the CIFAR-100 dataset. We provide the comparison results on four different optimizers: SGD, SGD with momentum, Adam, and Adagrad in Table 6.

**Table 6.** Validation accuracy (%) over different optimizer based on ResNet-110 trained on CIFAR-100.

| Optimizer | | +RALR | Acc. |
|---|---|---|---|
| SGD | - | × | $67.48 \pm 0.35$ |
| | | √ | $\mathbf{69.99 \pm 0.15}$ |
| | + momentum | × | $74.53 \pm 0.41$ |
| | | √ | $\mathbf{75.87 \pm 0.07}$ |
| Adam | | × | $72.55 \pm 0.06$ |
| | | √ | $\mathbf{73.55 \pm 0.09}$ |
| Adagrad | | × | $64.56 \pm 0.80$ |
| | | √ | $\mathbf{65.01 \pm 0.86}$ |

Specifically, we initially set the learning rate to 0.001 and decayed at the 150th epoch for optimizer Adam (besides, we also test the Adam with other hyper-parameters, but the network performance cannot be improved), and we initially set the learning rate to 0.1 and decayed at the 150th epoch for optimizer Adagrad. For optimizer SGD, the hyper-parameters are the sam as the CIFAR-100 in Section 5.1.

*5.3. RALR's Effectiveness on Other Techniques*

To verify that RALR is complementary to existing popular data augmentation and regularization techniques, we do some experiments with ResNet-56 on the CIFAR100 dataset. We compare our proposed method with four state-of-the-art methods: Cutout, Mixup, CutMix, and Label Smoothing.

As seen in Table 7, we observe that if we combine the techniques above with RALR methods, the classification performances are outperformed than when we use these methods alone.

Specifically, we select the hole size of $8 \times 8$ pixels for the CIFAR-100 dataset when training on the full datasets for Cutout, which is the best parameters adapted in [11]. For Mixup, we tested the hyper-parameter $\alpha = 1.0$. $prob = 0.5$ and $\alpha = 1.0$ are adapted in CutMix. We use $smoothing = 0.1$ for Label Smoothing technique.

**Table 7.** Validation accuracy (%) over different techniques based on ResNets trained on CIFAR-100.

| Method | +RALR | ResNet-56 | ResNet-110 |
|---|---|---|---|
| +baseline | $\times$ | $73.69 \pm 0.07$ | $74.53 \pm 0.41$ |
| | $\checkmark$ | $\mathbf{74.54 \pm 0.04}$ | $\mathbf{75.87 \pm 0.07}$ |
| +Cutout | $\times$ | $74.55 \pm 0.05$ | $75.34 \pm 0.12$ |
| | $\checkmark$ | $\mathbf{75.02 \pm 0.02}$ | $\mathbf{76.11 \pm 0.05}$ |
| +Mixup | $\times$ | $75.66 \pm 0.07$ | $77.89 \pm 0.05$ |
| | $\checkmark$ | $\mathbf{76.02 \pm 0.04}$ | $\mathbf{78.26 \pm 0.05}$ |
| +CutMix | $\times$ | $76.67 \pm 0.11$ | $77.71 \pm 0.13$ |
| | $\checkmark$ | $\mathbf{77.13 \pm 0.03}$ | $\mathbf{78.52 \pm 0.07}$ |
| +Label Smoothing | $\times$ | $74.05 \pm 0.11$ | $75.01 \pm 0.11$ |
| | $\checkmark$ | $\mathbf{74.94 \pm 0.07}$ | $\mathbf{75.98 \pm 0.04}$ |

*5.4. Ablation Study*

Effect of the random interval factor: We explore the effect of the random interval factor *low* and *high* according to the classification performance. We change the random interval factor setting *low* from 1 to 5 and *high* from 2 to 9; the probability to take RALR method is fixed to $p_r = 0.5$. In addition, we evaluate the classification performance of network ResNet-110 on the CIFAR-100 dataset using the above factors. Partial validation accuracy is demonstrated in Table 8. As shown in Table 8, with the increase of random factor *low*, the classification accuracy decreases after *low* $\geq$ 2. With the increase of random factor *high* when setting the random factor to *low* = 1, the accuracy raises and reaches the highest when *high* = 3, and then the accuracy decreases.

**Table 8.** Impact on parameters of different *low, high* with fixed probability of $p_r = 0.5$ when taking RALR method.

| Model | Low | High | Acc. |
|---|---|---|---|
| | 1 | 2 | $75.37 \pm 0.82$ |
| | 1 | 3 | $\mathbf{75.87 \pm 0.07}$ |
| | 1 | 5 | $75.35 \pm 0.22$ |
| | 1 | 7 | $70.17 \pm 1.58$ |
| ResNet-110 | 2 | 3 | $75.71 \pm 0.57$ |
| | 2 | 5 | $75.81 \pm 0.11$ |
| | 2 | 7 | $75.13 \pm 0.45$ |
| | 3 | 5 | $71.88 \pm 2.11$ |
| | 3 | 7 | $71.79 \pm 3.01$ |

Effect of the probability: Another important hyper-parameter of RALR is amplification probability $p_r$. We do some experiments to explore the influence of probability $p_r$ with a different threshold. We set amplification factor $p_r$ from 0.1 to 0.9, and evaluate the accuracy of network ResNet-110 on the CIFAR-100 dataset. As illustrated in Table 9, the accuracy reaches the highest at amplification probability $p_r = 0.5$, and variance reaches the lowest at the same time. The variance reaches the lowest and accuracy reaches the highest when $p_r = 0.5$, *low* = 1, and *high* = 3.

**Table 9.** Impact on parameters of probability with fixed *low* and *high* when taking RALR method.

| Model | Prob | Low | High | Acc. |
|---|---|---|---|---|
| ResNet-110 | 0.3 | 1 | 3 | 74.44 ± 0.87 |
| | | 2 | 5 | 73.98 ± 1.36 |
| | 0.4 | 1 | 3 | 75.31 ± 0.11 |
| | | 2 | 5 | 75.26 ± 0.23 |
| | 0.5 | 1 | 3 | **75.87 ± 0.07** |
| | | 2 | 5 | 75.81 ± 0.11 |
| | 0.6 | 1 | 3 | 75.53 ± 0.31 |
| | | 2 | 5 | 75.40 ± 0.62 |

## 6. Conclusions

In this work, we propose a new learning rate changing policy, where the global learning rate randomly amplifies at a certain probability between reasonable boundaries, obtaining a near-optimal network performance compared to the traditional monotonically decreasing way. The experimental results demonstrate that our RALR method achieves consistent improvements in various tasks and datasets. The RALR policy is easy to implement and, unlike adaptive learning rate methods, incurs essentially no additional computational expense. This work is limited by not entirely searching all pf the applications for the RALR methods yet. We expect to determine if similar policies work for training networks over different research domains in future work.

**Author Contributions:** Conceptualization, M.L. (Ming Liu) and J.D.; methodology, J.D.; validation, M.L. (Minghui Liu), T.X., and X.C.; formal analysis, J.D. and H.G.; writing—original draft preparation, J.D.; writing—review and editing, X.C. and X.W. All authors have read and agreed to the published version of the manuscript.

**Funding:** This research received no external funding.

**Institutional Review Board Statement:** Not applicable.

**Informed Consent Statement:** Not applicable.

**Data Availability Statement:** CIFAR-10 and CIFAR-100: http://www.cs.utoronto.ca/~kriz/cifar.html; CUB-200-2011: http://www.vision.caltech.edu/visipedia/CUB-200-2011.html; Standford Dogs: http://vision.stanford.edu/aditya86/ImageNetDogs/main.html; Standford Cars: http://ai.stanford.edu/~jkrause/cars/car_dataset.html; PASCAL VOC: https://pjreddie.com/projects/pascal-voc-dataset-mirror/; WMT 2014: https://www.statmt.org/wmt14/translation-task.html.

**Conflicts of Interest:** The authors declare no conflict of interest.

## Abbreviations

The following abbreviations are used in this manuscript:

| | |
|---|---|
| RALR | Random amplify learning rate |
| LR | Learning rate |
| SGD | Stochastic gradient descent |
| CAM | Class activation mapping |

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
