# Peer review of "RALR: Random Amplify Learning Rates for Training Neural Networks"

_applsci, doi:10.3390/app12010268_

Round 1

Reviewer 1 Report

Overall, the paper seems to be structured and written quite well. Extensive experimentation supports the final results which make it appealing to believe that this approach is quite promising. Yet, I still identified some possible issues (which may be explained by the authors, of course) in this approach

First, it is rather a heavily randomized technique with some control (limiting bounds). This is not necessarily a disadvantage - actually, swarm intelligence-based techniques make also extensive use of controlled randomization and often succeed in reaching very promising results. Hence, it would highly depend on the initialization of the initial vector. Moreover, Figure 1 shows very high fluctuations in training and validation loss which require a larger number of epochs (100-200 in this figure) for convergence. This may indicate that such an approach may not be a good option if fewer resources are available for training, and a smaller number of epochs might have to be used to achieve satisfiable performance (especially when larger training datasets or complex architectures are considered). Moreover, the distribution of amplifications might not be uniform (i.e., they may happen to be concentrated at the beginning or at the end) which would make the learning rate highly fluctuate. I suppose a smaller amplification probability makes it more stable, yet at the same time, it tends to go towards more conservative settings. Hence, multiple runs might be required to run optimal results (although the authors indicate in their results in Tables 1-8 that the variance is quite low, if I understood correctly, see my further comment)

Second, I did not find the baseline which is mentioned in Figure 1.

Third, what do +/- values represent in Table 1-8? Does it indicate that multiple runs were executed and the outcomes are averaged? How many runs were performed then (variable number of runs per experiment, etc.)? While running multiple experiments is the valid way to perform such experiments, this explanation is missing in the paper (or at least I failed to find them)

Reviewer 2 Report

1.abstract must include the numerical result

2.the author must include and summarize better if your proposed method outperformed other method, may be graphical result or so is needed.

3. conclusion should not include table, you should address problem statement shortly, methodology shortly and main result, discussion and research extension.
